# The Silent Threat to Women’s Fertility: Uncovering the Devastating Effects of Oxidative Stress

**DOI:** 10.3390/antiox12081490

**Published:** 2023-07-26

**Authors:** Aris Kaltsas, Athanasios Zikopoulos, Efthalia Moustakli, Athanasios Zachariou, Georgia Tsirka, Chara Tsiampali, Natalia Palapela, Nikolaos Sofikitis, Fotios Dimitriadis

**Affiliations:** 1Department of Urology, Faculty of Medicine, School of Health Sciences, University of Ioannina, 45110 Ioannina, Greece; a.kaltsas@uoi.gr (A.K.); kzikop@uoi.gr (A.Z.); azachariou@uoi.gr (A.Z.); nsofikit@uoi.gr (N.S.); 2Laboratory of Medical Genetics in Clinical Practice, Faculty of Medicine, School of Health Sciences, University of Ioannina, 45110 Ioannina, Greece; ef.moustakli@uoi.gr (E.M.); g.tsirka@uoi.gr (G.T.); 3Independent Researcher, 55131 Thessaloniki, Greece; x.tsiampali@gmail.com; 4Medical Faculty, Medical University of Sofia, 1431 Sofia, Bulgaria; nataliapalapela21@gmail.com; 5Department of Urology, Faculty of Medicine, School of Health Sciences, Aristotle University of Thessaloniki, 54124 Thessaloniki, Greece

**Keywords:** oxidative stress, endometriosis, polycystic ovary syndrome, diminished ovarian reserve, antioxidants

## Abstract

Oxidative stress (OS), which arises through an imbalance between the formation of reactive oxygen species (ROS) and antioxidant defenses, plays a key role in the pathophysiology of female infertility, with the latter constituting just one of a number of diseases linked to OS as a potential cause. The aim of the present article is to review the literature regarding the association between OS and female infertility. Among the reproductive diseases considered are endometriosis and polycystic ovary syndrome (PCOS), while environmental pollutants, lifestyle variables, and underlying medical conditions possibly resulting in OS are additionally examined. Current evidence points to OS likely contributing to the pathophysiology of the above reproductive disorders, with the amount of damage done by OS being influenced by such variables as duration and severity of exposure and the individual’s age and genetic predisposition. Also discussed are the processes via which OS may affect female fertility, these including DNA damage and mitochondrial dysfunction. Finally, the last section of the manuscript contains an evaluation of treatment options, including antioxidants and lifestyle modification, capable of minimizing OS in infertile women. The prime message underlined by this review is the importance of considering OS in the diagnosis and treatment of female infertility. Further studies are, nevertheless required to identify the best treatment regimen and its ideal duration.

## 1. Introduction

Millions of women worldwide struggle with female infertility, which is a serious public health problem. Oxidative stress (OS) has been identified as a major factor in the pathophysiology of female infertility. In OS, reactive oxygen species (ROS) and antioxidants are out of balance, resulting in cellular damage. An excess of pro-oxidants may cause OS when their systemic expression surpasses the capacity of a biological system to rapidly detoxify the reactive intermediates or to repair the damage they have caused [1]. This mechanism may be altered if ROS, reactive nitrogen species (RNS), antioxidant defense mechanisms, or else any combination of the latter arise [2]. In the case that all molecules are reconverted to their reduced state after oxidation, a certain degree of ROS is needed to enable normal cellular functioning [3]. However, in women, this antioxidant defense mechanism may be overwhelmed due to excessive formation of ROS, leading to an environment unable to support normal physiological processes [4]. In other words, when the antioxidant system is exhausted by an excess of ROS, the female genital tract will be negatively affected. Oocytes and follicles are damaged by OS in the female reproductive system, which impairs implantation, alters endocrine function, and damages the endometrium. The result is an alteration in ovulation, steroidogenesis, and oocyte maturation, which in turn hastens the natural process of apoptosis in granulosa cells. Some of the issues that may arise from the above condition are endometriosis, polycystic ovarian syndrome (PCOS), and unexplained infertility [3], while other possible consequences are such pregnancy complications as preterm birth, hypertension, and intrauterine growth restriction (IUGR) [5]. Oxidative abnormalities are associated with malnutrition, obesity, and negative lifestyle behaviors, including alcohol consumption, smoking, and recreational drug use [3,6] while being exposed to ovotoxins in the workplace and the environment can also adversely affect fertility [7,8,9]. Infertile couples frequently resort to assisted reproductive technology (ART) to increase their chances of conception [3], while there are ongoing investigations into the addition of antioxidants to the culture medium as a means of improving the success rate [10]. This article examines the ways in which OS adversely affects the female reproductive system while presenting a number of remedies that can halt or reduce the damage.

## 2. ROS in the Female Reproductive System

### 2.1. Source of ROS Production in the Female Reproductive System

The main components of the female reproductive system are the ovaries, fallopian tubes, uterus, and cervix. Potential sources of ROS in the female reproductive system are granulosa cells, oocytes, cumulus cells, and endometrial cells [11]. Granulosa cells play an important role in the maturation and development of the oocyte by surrounding it and it is during oocyte development and ovulation that these cells form ROS. ROS, which are defined as oxygen free radicals, are produced as intermediates during the metabolic process [12]. Among the ROS produced by granulosa cells during follicular growth and that play a part in controlling ovulation and corpus luteum activity, are hydrogen peroxide (H_2_O_2_), superoxide anions (O^2−^), and nitric oxide (NO) [13]. Because they contain many mitochondria, these being the main generators of ROS, oocytes are particularly vulnerable to OS. ROS are among the by-products of the electron transport cycle during ATP generation in mitochondria [14]. In addition, oocytes generate ROS during meiosis, this being an essential process for oocyte development and fertilization. During ovulation, cumulus cells form in a circle around the oocyte, while during both ovulation and fertilization, these cells release ROS. Laboratory studies have shown that during ovulation, the cumulus cells secrete H_2_O_2_ and O^2^ which regulate fertilization and implantation [15]. The endometrial cells which line the uterus play a crucial role in fertilization and implantation, while during menstruation and pregnancy they produce ROS. It has been demonstrated that endometrial cell production of H_2_O_2_ and O_2_ during the menstrual cycle contributes to the control of endometrial receptivity and implantation [16,17].

### 2.2. The Physiological Role of OS in Female Reproduction

ROS set off a chain reaction that involves membrane lipids, DNA, RNA, proteins, carbohydrates, and other macromolecules in the cell [18]. Cell damage can be attributed to OS caused by a cascade of lipid peroxidation process, DNA damage, membrane disruption, protein formation disorders, and loss of adenosine triphosphate energy [19]. During aerobic respiration, the mitochondrial electron transport chain, endoplasmic reticulum, and nuclear membrane generate endogenous ROS [20]. Moreover, the formation of ROS is also influenced by certain metabolic processes, including the activities of xanthine oxidases, cytochrome P450, and nicotinamide adenine dinucleotide phosphate oxidases [20]. Higher levels of ROS are produced in response to psychological stress as well as to drug, alcohol, and tobacco use and inactive lifestyle [21]. OS is caused by ROS, which includes superoxide anion, hydrogen peroxide, hydroxyl radical, peroxyl, and hydroperoxyl [12]. Proliferation, differentiation, and apoptosis are just some of the cellular biological activities regulated by the expression of specific genes and proteins [22,23]. ROS play a role in the menstrual cycle by influencing tissue remodeling, hormone signaling, and cyclic endometrial changes in the female reproductive system. ROS in the ovary regulate ovarian steroid hormone production, follicle synthesis, maturation, ovulation, and tubal function [24]. Corpus luteum breakdown, implantation, and the normal birth process are all affected by ROS [24,25]. ROS and antioxidants in the ovaries, follicular fluid, and peritoneal fluid all play a role in oocyte quality, oocyte fertilization, implantation, and embryo development [26]. Several in vivo and in vitro studies have shown that ROS are involved in angiogenesis-promoting vascular endothelial growth factor signaling [27]. They also suggest that ROS play a role in folliculogenesis and early embryonic development. ROS and antioxidant levels regulate the microenvironment of the follicular fluid, which process determines which follicle becomes dominant. ROS can trigger ovulation in mature Graaf follicles, and any change in this ROS level can disrupt ovulation [28]. Increased catalase, glutathione, and estrogen production in response to ROS prevents apoptosis in the mature Graafian follicle during ovulation [3].

### 2.3. Mechanisms by Which OS Damages the Female Reproductive System

Cells are damaged by OS in many ways, including lipid peroxidation, protein oxidation, DNA damage, and mitochondrial dysfunction [29]. Consequences of the destruction by OS of oocytes and follicles in the female reproductive system include defective implantation, altered endocrine function, and endometrial damage. Lipid peroxidation is a process via which lipids are degraded by oxygen, thus producing lipid peroxides [30], while lipid peroxidation-induced damage to the cell membrane weakens the viability and function of oocytes and follicles within the female reproductive system. Moreover, OS and cell damage are exacerbated by proinflammatory cytokine formation that has also been induced by lipid peroxidation. Oxidization of amino acid residues in proteins results in the disruption of the original functions of the proteins, while protein oxidation diminishes the viability and function of proteins in the developing oocytes and follicles within the female reproductive system [31]. Protein oxidation, while causing OS and cell damage, additionally releases proinflammatory cytokines. Oxidative alteration of DNA leads to its damage, resulting in dysfunctional DNA and mutations [32], while in the female reproductive system, DNA damage negatively impacts the viability and function of oocytes and follicles by damaging their DNA. DNA damage, by increasing OS and cell damage, activates the production of proinflammatory cytokines. Meanwhile, energy production is reduced when mitochondrial activity is impaired so that ROS are released at elevated levels. In the female reproductive system, mitochondrial dysfunction damages the viability and function of oocytes and follicles [33], whereas OS and cellular damage are intensified due to the production of proinflammatory cytokines induced by mitochondrial dysfunction (Figure 1).

### 2.4. What Are the Adverse Effects of OS on the Female Reproductive System?

#### 2.4.1. Oocyte Quality

The degree of viability of the oocyte depends greatly on its quality, and this quality may be reduced due to oxidative damage to DNA, lipids, and proteins caused by OS [34]. Meanwhile, chromosome abnormalities and problems with conception may arise if there is ROS-induced DNA damage, including double- and single-strand breaks and nucleotide alterations [35]. Lipid peroxidation, also potentially caused by OS, reduces fluidity and permeability of the oocyte membrane and, therefore, its ability to be fertilized [36]. OS can also cause protein oxidation, which may impede oocyte development and, therefore, fertility [37] (Figure 1).

#### 2.4.2. Embryonic Growth

Embryonic development is a highly complex process capable of being affected by many factors, one being OS, which can obstruct embryo development by altering gene expression: This can generate mitochondrial dysfunction which, in turn, triggers DNA damage [38]. ROS-induced DNA damage can even stop embryonic development by producing chromosomal abnormalities. It is of note that diminished ATP generation and stunted embryo development have been associated with OS [39], while, moreover, OS may modify gene expression, causing cells to be more susceptible to death thus aggravating aberrant development [40] (Figure 1).

#### 2.4.3. Implantation

Implantation is a process that is crucial to establishing a healthy pregnancy. However, due to its negative impact on endometrial inflammation, trophoblast invasion, and gene expression, OS can play a role in implantation failure [41]. Notably, by altering the embryo’s gene expression, OS can trigger aberrant development, this leading to a greater risk of apoptosis [12]. The establishment of pregnancy is furthermore dependent on trophoblast invasion, which is also susceptible to hindrance by OS. Meanwhile, ROS-induced endometrial inflammation may diminish the capacity of the endometrium to receive and nourish the embryo [42] (Figure 1).

## 3. Factors Affecting Female Fertility Associated with OS

### 3.1. Age

Given that fertility decreases with age, maternal age is naturally a big factor in infertility. For example, by age 44, the woman experiences reduced estrogen levels and diminished protection from oxidative damage to the endometrium [43]. Hormone replacement therapy (HRT) is able to defend against OS by preventing the effects of the age-related reduction in antioxidant levels, while it may be additionally capable of considerably slowing down loss in infertility, although more studies are needed to support this hypothesis [44]. In addition to maternal age, paternal age also plays a role in fertility, as aging negatively affects gamete and semen quality and also causes oxidative DNA damage (Figure 1). In a word, the older the father, the more greatly sperm DNA is exposed to OS [45].

### 3.2. Body Weight

(i) Obesity: Obesity poses significant threats to female fertility by interfering with hormonal regulation and menstrual cycles [46,47,48]. The disease’s pathology includes the overproduction of ROS, leading to OS that damages reproductive cells and tissues in women’s reproductive systems [46,49]. ROS production from obesity can interfere with vasodilation and blood flow to reproductive organs, leading to fertility issues [49]. Furthermore, obesity often triggers hormonal imbalances that alter ovulation regularity leading to conditions like PCOS which is the leading cause of female infertility [47,48]. The mechanisms underlying the association between obesity and hormonal disturbances are complex and multifactorial, involving alterations in adipokine secretion, insulin resistance, and inflammation [50]. Furthermore, obesity-induced OS can directly damage female fertility by damaging oocytes and impairing the functioning of the hypothalamic-pituitary-ovarian axis [47]. Therefore, understanding the complex interactions among obesity, hormone disruption and fertility outcomes for obese women is vital in order to increase fertility outcomes.

(ii) Underweight: Malnourished reproductive women have impaired endothelium-dependent vasodilation, which in turn causes OS [51].

### 3.3. Lifestyle Factors

(i) Cigarette smoking: It is well known that smoking during pregnancy increases the risk of infertility, pregnancy problems, fetal loss, fetal developmental delay, preterm birth, and miscarriage [52]. Toxic compounds and prooxidants in cigarettes cause the body to release ROS, leading to OS in the microenvironment of follicles [53].

(ii) Alcohol usage: Alcohol consumption produces metabolites, such as acetyl and methyl radicals, which are responsible for the formation of ROS. Alcohol consumption during pregnancy increases ROS in maternal plasma, causes lipid peroxidation, and decreases antioxidant activity and glutathione (GSH) levels of superoxide dismutase (SOD). Therefore, alcohol consumption during pregnancy can lead to IUGR, preterm birth, low birth weight, increased risk of congenital diseases, miscarriage, and prematurity [3].

(iii) Recreational Drug usage: Tetrahydrocannabinol, the active ingredient in marijuana (recreational drug), generates free radicals that can affect both the central and peripheral nervous systems. Delta-9-tetrahydrocannabinol (THC) is another major substance constituent that produces psychological effects in smokers. The induction of DNA damage by this THC has been linked to the production of ROS [54]. Just as nicotine causes OS and lipid peroxyl radicals via its metabolites, cocaine does so via its metabolites. Norcocaine, another oxidative metabolite of cocaine, generates OS by depleting GSH reserves [55], while formaldehyde (an oxidative metabolite of cocaine) generates ROS. In addition, ROS causes apoptosis [56]. Research indicates that THC, the primary psychoactive component of cannabis, can adversely impact oocyte maturation and early embryonic development. Specifically, exposure to THC reduces the likelihood of oocytes reaching metaphase II and causes lower cleavage rates post-fertilization. Additionally, while no notable changes are seen in spindle morphology, there is an increase in apoptosis levels within the derived blastocysts, suggesting a disruptive effect of cannabis on reproductive processes [57].

### 3.4. Environmental and Occupational Exposures

(i) Pesticides: Wives of male agricultural workers who come into close contact with pesticide compounds such as DDT (organochlorine insecticides) have an increased risk of miscarriage or spontaneous abortion according to extensive research [58]. Polychlorinated biphenyls, often referred to as PCBs (a pesticide), are known to increase free radical production by inducing OS through endothelial cell dysfunction and cell membrane disintegration. Vitamin E levels decrease after PCB exposure [59], while exposure to organophosphate pesticides leads to OS due to decreased GSH and increased ROS [60].

(ii) Endocrine-Disrupting Chemicals: The male reproductive system is negatively affected by endocrine-disrupting chemicals (EDCs), including phthalates, which also contribute to the development of OS. In addition, elevated concentrations of phthalate metabolites in urine have been found in workers in the polyvinyl chloride industry [61]. Phthalate metabolites in urine have been observed to cause sperm apoptosis and ROS formation, even after controlling for age, smoking status, and coffee consumption. Evidence shows that bisphenol A (BPA) can cause OS, reducing a man’s ability to have children. Chemicals like phthalates and BPA, which disrupt reproductive hormones, have been associated with fertility challenges in women, including repeated miscarriages and unexplained infertility. The risk appears to be exacerbated among individuals engaged in commerce-related occupations due to heightened exposure, thereby indicating a potential occupational hazard for reproductive health [62].

## 4. OS Has Been Implicated in a Variety of Reproductive Disorders

### 4.1. Polycystic Ovarian Syndrome

PCOS, one of the main endocrine disorders of reproductive-aged women affecting approximately 18% of women of this age group, is characterized by hyperandrogenism, inability for normal folliculogenesis, and polycystic ovaries [63], while menstrual irregularities, e.g., absent or heavy periods, constitute typical PCOS symptoms. As a result, 90% of PCOS women suffer from anovulatory infertility. PCOS is moreover often characterized by insulin resistance, this disorder probably being a major factor in the development of the condition. Insulin resistance arises when the cells cease to respond to the hormone insulin, which is vital to controlling blood sugar levels: This results in an elevation of androgen production in the ovaries, which, in turn, disrupts the regular menstrual cycle and causes the development of ovarian cysts.

The presence of insulin resistance can make weight reduction more challenging, thus exacerbating PCOS symptoms. Furthermore, it has been shown that insulin resistance generates chronic low-grade inflammation, which is known to play a part in PCOS development. An additional consequence of insulin resistance is the generation of ROS. Due to the high blood glucose levels, ROS is generated, which causes oxidative damage [64]. The association of PCOS with OS is likely due to the low levels of antioxidants found in this condition [65], while mitochondrial dysfunctions in PCOS patients are mainly attributable to patients’ reduced oxygen (O_2_) consumption, lower GSH levels, and elevated formation of ROS [66]. In PCOS women, the above inflammatory state is characterized by increased mononuclear cell production [67], presumed to be an outcome of an exaggerated response to hyperglycemia and C-reactive protein (CRP).

### 4.2. Endometriosis

Endometriosis, a benign, estrogen-dependent, chronic gynecologic disorder, affects 6–10% of reproductive-aged women and can manifest via pelvic discomfort and infertility. It is specifically characterized by the development of endometrial tissue in sites other than the uterus. The most frequent areas where endometriosis lesions occur are the ovaries and pelvic structures. Rarely, the lesions can also impact the lungs, the abdomen, and the viscera. The exact causes of endometriosis, which appear to be multifaceted and complex, remain uncertain [68]. It is thought to arise from a combination of factors, potentially including inflammation, a weakened immune system, and genetic predisposition [69].

As concerns pelvic endometriosis, the most plausible explanation to date for its cause is retrograde menstruation and implantation [70]. Although research results regarding the detection of OS markers in endometriosis patients are to date conflicting [71], a number of studies have observed elevated levels of OS markers in endometriosis patients [72,73,74,75,76]. However, others have reported no evidence of increased OS in the peritoneal fluid or bloodstream of patients [77,78,79]. Of note, endometriosis cysts with frequent cyclic hemorrhage are often observed to have higher free iron levels than other ovarian cysts, while endometriosis cysts showing increased levels of lipid peroxides, 8-OHdG, and antioxidant indicators point to the presence of OS and DNA damage, but also to an elevated antioxidant response, the latter results demonstrating that endometrial cysts display an altered redox state [80]. Since iron promotes the production of ROS and DNA damage, several treatments are recommended to counter this effect. Patients with endometriosis have less likelihood of getting pregnant than women without the disorder. It is suspected that in endometriosis patients, low oocyte and embryo quality and spermatotoxic peritoneal fluid induced by ROS could contribute to subfertility [81]. Moreover, low concentrations of ascorbic acid [75] and of glutathione peroxidase (GPx) [74] have been observed in the peritoneal fluid of these women, while lower GPx concentrations have been linked to decreased progesterone response within endometriosis cells [82].

While endometriotic cells are observed to contain large amounts of ROS, the origin of ROS is yet unknown. It is hypothesized that elevated cell proliferation and prevention of apoptosis in endometriotic cells may be correlated with impaired detoxification systems, which partially induce excess ROS and OS. There is a need for further examination of dietary and supplemental antioxidant intake in different populations to establish whether antioxidant status and/or intake can play a part in the development, maintenance, or curing of endometriosis [71].

### 4.3. Unexplained Infertility

Unexplained infertility is characterized when couples have engaged in unprotected intercourse for at least 12 months without successful conception, despite standard fertility evaluations revealing no identifiable causes for infertility. While research indicates that an imbalance of ROS and antioxidant defenses may contribute to its pathogenesis, it is critical to recognize this is only one plausible explanation [83]. Other potential causes could include peritoneal endometriosis, subtle tubal lesions, and chronic endometritis [84,85]. Another noteworthy potential explanation is a path anomaly in the methyl-tetra-hydrofolate reductase (MTHFR) gene, which is involved in folate metabolism and, consequently, DNA, lipid, and protein methylation. This polymorphism can disrupt homocysteine levels and homeostasis, potentially affecting oocyte quality and endometrial development [86]. However, the precise role of these factors, including the potential benefits of antioxidant supplementation, requires further investigation.

## 5. Pregnancy Complications

### 5.1. The Placenta

The placenta is a vital prenatal organ enabling the transport of oxygen, nutrients, and hormones between mother and fetus, while it additionally protects and immunizes the developing fetus, trophoblastic invasion of the maternal spiral arteries being the initiator of these placental activities [5]. There are indications of morphological adaptation to hypoxia in the placenta, while modifications to the placental vasculature occur to ensure sufficient blood flow between the mother’s and the fetus’s systems. A number of placental disorders, including chronic villitis and maternal or fetal vascular malperfusion, lower the levels of oxygen that are exchanged between mother and fetus. Clinically speaking, several indicators of OS are observed, such as MDA and reduced levels of thiols. Physiological hypoxia is caused by the low O_2_ tension observed in early pregnancy before the trophoblastic plugs dissipate [87]. The syncytiotrophoblast, having no antioxidants at this stage, can easily be damaged by free radicals. In parallel to a significant increase in O_2_ tension [88], the formation of full maternal arterial circulation to the placenta occurs, which, however, is associated with an increase in ROS, eventually leading to OS [89]. ROS stimulates both the proliferation of cells and the expression of genes in physiological quantities [90]. Towards the end of the first trimester, in response to increased O_2_ tension and OS, the placenta upregulates antioxidant gene expression and activity, thereby protecting fetal tissue against the damaging effects of ROS during all the critical stages of development and organogenesis [91] (Figure 1). Antioxidants found in the placenta include heme oxygenase (HO)-1 and -2, Cu, Zn-SOD, catalase, and GPx [92]. If placental OS begins prematurely, the syncytiotrophoblast may degenerate, given that maternal blood flow will in this case reach the intervillous gap too soon. Some of the possible adverse outcomes of this state are miscarriage [88,93,94], recurrent pregnancy loss (RPL) [95], and preeclampsia [96].

### 5.2. Spontaneous Abortion

Spontaneous abortion denotes the non-induced loss of pregnancy before 20 weeks of gestation. Almost one-half of all miscarriages can be attributed to chromosome anomalies, which point, to a large degree, to the underlying cause of the disorders. Additional reasons are maternal factors, including uterine anomalies, congenital disabilities, various diseases and infections, and idiopathic causes [97]. One theory attributes spontaneous abortion mainly to high placenta OS. As described above, frequently between the 10th and 12th week of pregnancy, an oxidative burst occurs in the placentas of healthy pregnancies: the elevated antioxidant activity results in the OS returning to its baseline as placental cells steadily adjust to their new oxidative environment [88]. By 8–9 weeks of gestation, miscarriages will be seen to differ from regular pregnancies [88,94], with high levels of heat shock protein 70 (HSP70) and nitrotyrosine along with indicators of apoptosis being observed in the villi of such placentas: all this strongly indicates oxidative damage to the trophoblast, resulting in subsequent loss of pregnancy [91]. The production and activity of antioxidant enzymes increase with gestational age, they are at this stage unable to counteract rises in ROS. Premature onset of OS has been associated with impaired placental development and/or accelerated syncytiotrophoblast degeneration, resulting in miscarriage [97].

Of interest, a link has been identified between deficiency in selenium (Se) and spontaneous abortion [98,99] and RPL loss [99], this is possibly due to a decrease in GPX’s detoxifying abilities, which occurs in the setting of Se deficiency. Early pregnancy loss has also been associated with a decline in biomarker serum prolidase activity, a marker of the turnover of extracellular matrix and collagen. Moreover, serum prolidase levels have been inversely correlated with better OS, possibly explaining the higher placental vascular resistance and endothelial dysfunction seen in patients with diminished and dysregulated collagen turnover [100]. It was observed that women who experienced a miscarriage during the early stages of pregnancy exhibited decreased levels of the antioxidant enzyme serum paraoxonase/arylesterase, while the same individuals also showed high vulnerability to lipid peroxidatin, this indicated by a negative connection with lipid hyperoxide [101]. OS-induced inflammatory mechanisms may, moreover, lead to apoptosis of placental tissues. Though the cause has not been clarified, the excess or inadequate release of maternal blood flow to the intervillous area has been linked to spontaneous and RPL to prevent miscarriage during the first trimester, several studies have been carried out using antioxidant supplements to restore depleted levels and counter an excessively oxidative environment. However, while antioxidant supplements could alleviate the problem, a meta-analysis of the relevant studies failed to offer any support for the above theory [102].

### 5.3. Recurrent Pregnancy Loss

RPL, which affects 1–3% of all pregnancies, is defined as three or more consecutive miscarriages. The causes of RPL may be found in only 50% of cases, since, notwithstanding findings indicating the involvement of OS in the pathogenesis of RPL [103], the remainder of cases lack any clear explanation [104,105]. Irregular placentation, which results in syncytiotrophoblastic degradation and OS, is thought to be a major contributor to idiopathic RPL [96], perhaps accounting for the increased susceptibility of the syncytiotrophoblast to OS during the first trimester. In line with this hypothesis, it has been observed that patients with RPL have increased plasma lipid peroxides and GSH along with decreased vitamin E and beta-carotene [106]. Furthermore, plasma GSH levels were found to be significantly elevated in women with a history of RPL, this pointing to a reaction to elevated OS [107]. On the other hand, although individuals with idiopathic RPL were reported in one study to have higher amounts of MDA, another research study demonstrated that their antioxidant enzymes, GPx, SOD, and catalase levels were considerably lower than the average. Yet other researchers have reported a correlation between polymorphisms in antioxidant enzymes and increased incidence of RPL [108,109,110], while some studies have linked certain null genetic variants of GST enzymes to RPL [111,112,113].

Amin et al. have observed that N-acetylcysteine (NAC) and folic acid enhances pregnancy outcomes in women with unexplained RPL [114], proposing that, by decreasing oxidative genotoxicity, inhibiting the release of proinflammatory cytokines, and preventing endothelial apoptosis, NAC may be able to moderate OS-induced reactions and processes which cause oxidative damage during complicated pregnancies [115]. It is hypothesized that the intake of antioxidants could facilitate the restoration of antioxidant defenses as well as the mitigation of placental apoptosis and inflammatory responses that are related to severe OS. NAC is a particularly attractive potent antioxidant given that it contains numerous sulfhydryl groups, while, thanks to its thiol properties, it is capable of increasing GSH levels in cells as well as directly scavenging free radicals [116,117].

### 5.4. Preeclampsia

Even in the absence of hypertension before pregnancy, women may develop preeclampsia, thus facing the risk of serious complications. The fact that the incidence of preeclampsia is 3 to 14% of pregnancies means that it is a major source of maternal and fetal morbidity and mortality [118,119]. In women with early-onset preeclampsia, indicators of placental OS include protein carbonyls, lipid peroxides, nitrotyrosine residues, and DNA oxidation [89,120]. It is hypothesized that preeclampsia is caused by inadequate spiral artery conversion, which prevents placental perfusion resulting in moderate damage to ischemia-reperfusion [121,122,123]. Alterations in gene expression observed in preeclampsia, which could be attributed to ischemia-reperfusion damage in trophoblastic and endothelial cells [124], might explain the association of preeclampsia with low birth weight and inadequate implantation [5]. However, it is important to clarify that ovarian stimulation, a procedure common in ART, is not a universal cause of preeclampsia in all pregnancies. While ovarian stimulation may increase the risk of preeclampsia in ART pregnancies possibly due to hormonal changes, it does not necessarily contribute to preeclampsia in spontaneous pregnancies [125].

Preeclampsia has been associated with excessive apoptosis of villous trophoblasts, ovarian stimulation being proposed as a likely cause. The maternal blood of women with preeclampsia contains microparticles of syncytiotrophoblast microvillus membrane (STBMs), which have been reported to induce endothelial cell damage in vitro [126].

To identify the presence of placental OS, quantitative determination of elevated levels in the blood of ROS, including hydrogen peroxide [127], or lipid peroxidation indicators, such as MDA [118,128,129,130], or thiobarbituric acid reactive substances (TBARS), is carried out [118,127]. Since preeclampsia has been related to increased vasoconstrictor H_2_O_2_ and reduced levels of the vasodilator NO, this might account for the vasoconstriction and hypertension observed in this disorder. On the other hand, some studies have reported elevated NO levels in the bloodstream [131,132] and the placenta [133], while other research has shown that endothelial cell damage occurs in preeclampsia patients due to increased generation of the SO anion and lower NO release because of neutrophil regulation [134].

Trophoblasts, vascular endothelial cells, and numerous other cells depend on NAD(P)H oxidases to generate the SO anion. Activation of the enzymes, resulting in increased generation of SO anion, is associated with the pathophysiology of a number of vascular disorders [135]. The increase may take place via several physiological pathways, including the following. Angiotensin receptor (AT1) antibodies, particularly those directed against the second loop (AT1-AA) [136], can boost ROS production by stimulating NAD(P)H oxidase. Activated placental NAD(P)H produces more SO anion between weeks 6 and 8 of pregnancy than at term [137]; hence, NAD(P)H oxidase-mediated altered gene expression [23,138] can affect early placental development by means of dysregulated vascular formation and function. Higher NAD(P)H expression and ROS production have been observed in preeclamptic women than in those without the disorder [139]. It has also been reported that women who develop preeclampsia at a younger age generate more of the SO anion than those who are affected later [137]. Although the above findings shed some light on the part played by OS in the pathophysiology of placental malfunction in such reproductive conditions as preeclampsia, the precise mechanism of placental NAD(P)H activation is yet to be elucidated.

As has been demonstrated by Baker et al. [140], preeclamptic patients display elevated levels of paraoxonase-1 (PON-1), a finding that is in accordance with the role of OS in preeclampsia pathophysiology [140]. Preeclampsia patients have moreover exhibited elevated levels of PON-1 during mid-pregnancy, this apparently representing a protective mechanism against the potentially harmful impact of high OS seen in this syndrome [140]. In contrast, it has been reported that patients with severe preeclampsia and those presenting with clinical symptoms have significantly lower PON-1 [141,142].

Affected women have also been shown to display poor total antioxidant status (TAS), placental GPx, and depleted vitamins C and E [118,128,143]. A number of research studies have demonstrated that preeclampsia risk may be lowered in normal-weight or underweight women who have additionally been prescribed multivitamins during the entire periconceptional period [144,145], while other researchers have reported a higher risk among women with C deficiency [146]. On the whole, however, most authors show no reduction in preeclampsia incidence achieved via chronic antioxidant supplementation during pregnancy [102,147,148].

### 5.5. Intrauterine Growth Restriction

Most cases of IUGR, also known as a failure of the fetus to develop to its genetic development potential or fibroblast growth factor receptor, are attributable to complications involving the mother, the fetus, or both. The most common cause is uteroplacental dysfunction, denoting reduced maternal placental blood flow to the fetus. A number of studies have linked inadequate development of the spiral arteries to damage of placental ischemia/reperfusion. Metabolic activity along with cell development and proliferation produces ROS and OS; thus, the above process requires a high energy supply. Meanwhile, the chorioallantoic villi may be damaged by stimuli or mediators, possibly resulting in an inadequate trophoblastic invasion of the spiral arteries. OS is one of the chief triggers or mediators. It is, therefore, evident that insufficient spiral artery development can lead to ischemia/reperfusion, the latter exacerbating OS and contributing to the deterioration of placental tissue [149]. Indeed, in women with IUGR, increased free radical activity and indicators of lipid peroxidation are observed [150], while it has also been reported that these patients display reduced levels of antioxidant concentrations in plasma, placenta, and umbilical cords compared to controls, together with increased levels of MDA and xanthine oxidase [149]. It has additionally been shown that at 12 and 28 weeks of pregnancy, the DNA oxidation marker 8-oxo-7,8-dihydro-2-deoxyguanosine (8-oxodG) is significantly elevated in pregnancies complicated by growth-restricted fetuses as compared to a control group [150]. Ischemia and reperfusion injuries are major sources of ROS and OS. Reports show that hypoxia-reoxygenation [151] induces greater apoptosis in villous trophoblasts than does hypoxia alone [152]; also, that p53 [149], which regulates apoptosis, is highly upregulated in response to hypoxia in villous trophoblasts [152,153,154]. The severity of OS in IUGR placentas is exacerbated by low protein translation and signaling [155]. Since complex cellular mechanisms vulnerable to intracellular and extracellular stimuli compose fetal growth and development, it is plausible that cellular stress and programmed cell death play a part in inducing metabolic disease in adult IUGR offspring. While cell stress and death can often be protective processes, they can also be damaging, aiding in the development of metabolic disorders. Several studies show that an adverse perinatal environment may produce cell stress and cell death in the placenta, this impeding embryonic growth. This process may furthermore lead to OS mitochondria malfunction, endoplasmic reticulum (ER) stress inflammation, apoptosis, and autophagy in metabolic organs.

### 5.6. Preterm Labor

Globally, preterm births (PTBs), meaning those occurring before 37 weeks of gestation, are the leading cause of neonatal morbidity and mortality, affecting a total of 5–12% of all births. Traditionally, term births and preterm labor are regarded as similar processes following the same pathway. Romero et al. (2006) use the term “syndrome” to pinpoint the possible pathogenic causes of the commencement of preterm labor, although what are the specific causes and precipitating mechanisms of preterm labor are to date unknown [156]. PTB is physically challenging for neonates, given that they must adapt to a new environment for which they are not physiologically prepared, in particular as regards oxygenation and feeding. They will also experience an imbalance between the synthesis of oxidants and antioxidants which may induce OS: since the latter includes high concentrations of ROS, it can produce oxidative damage. Possible causes of elevated OS in preterm neonates are the following: oxygen resuscitation, preterm nutrition, blood transfusions, phototherapy inflammation and infection, increased metabolic rate, and immune antioxidant system. The majority of spontaneous preterm births are associated with several underlying conditions, which might include infection or inflammation, maternal or fetal stress, decidual hemorrhage indicative of placental abruption, or conditions that lead to uterine overdistension such as multiple pregnancies or polyhydramnios. It is also notable that endocrine disorders, issues related to the cervix such as incompetency, and poor vascular supply to the uterus (ischemia) could contribute to preterm labor. Understanding the specific cause in each case could be challenging, as these conditions often overlap and can simultaneously contribute to the onset of preterm birth [156]. While there are several potential causes of premature delivery, infection and inflammation in the uterus are considered to be the most likely [157]. A woman may undergo a number of procedures simultaneously. Certain women may be affected by genetics and inflammatory responses that constitute risk factors for preterm labor. Mustafa et al., comparing the maternal blood of women with preterm labor to that of women with term labor, observed notably more significant MDA and 8-OHdG and considerably lower levels of GSH [158]. Their study thus suggested that antioxidant capacities are decreased in preterm labor women, making them more susceptible to OS-induced damage.

Moreover, it has been demonstrated that women with preterm labor have exhibited lower activity of FRAP, an assay that measures the capacity to overcome oxidative damage and GST [158,159,160,161]. The latter results further support the belief that both mother and neonate are more susceptible to ROS-induced harm in an environment with high levels of OS and low antioxidant capacity. Both chorioamnionitis and histopathological infection have been implicated in preterm labor, while a number of studies have determined that preterm mothers’ elevated expression of Mn-SOD mRNA in the fetal membranes is linked to this phenomenon [162]. Preterm labor is associated with increased levels of OS levels and inflammation, and the higher expression of Mn-SOD mRNA observed in these cases may comprise a protective response to the described conditions. The amnion and choriodecidua of patients experiencing preterm labor have, for example, been shown to have notably more significant levels of the proinflammatory cytokines IL-1 beta, IL-6, and IL-8 as compared to those of women with spontaneous term labor [163]. Lower total antioxidant status (TAS) levels noted in women experiencing preterm labor than in those with uncomplicated pregnancies at a similar gestational age may point to increased OS [164]. It has also been demonstrated that PON1 activity is much lower in women who give birth prematurely than in controls [165]. The latter study suggests that the risk of preterm delivery may be higher in a pro-oxidant environment, which is caused by increased lipid peroxidation and diminished antioxidant activity of PON1.

It has additionally been reported that preemies’ GSH levels compared to those of offspring of full-term mothers are significantly lower [166]. Low Se levels in pregnant women’s blood have also been linked to preterm delivery [167]. Moreover, GST polymorphism was considerably higher in women who had experienced premature labor, pointing to a greater risk of oxidative damage. Finally, a maternal illness can cause OS, the resulting diminished antioxidant defenses being likely to increase the risk of premature birth.

### 5.7. Ectopic Pregnancy

The fallopian tubes play a critical role in female fertility by assisting in the movement of eggs from the ovaries to the uterus and serving as a site for fertilization. However, several factors, including oxidative stress, can impair the function of the fallopian tubes, which can have severe consequences for fertility and increase the risk of ectopic pregnancy. Coordination between smooth muscle contractions and cilia is essential for the egg’s passage through the fallopian tubes [168]. The egg is carried into the uterus by a directional flow produced by the beating of the cilia. Oxidative stress can interfere with the normal function of cilia, making tubal transfer difficult. Elevated ROS levels may impair the frequency of cilia beating and reduce the effectiveness of oocyte migration. As a result, altered fallopian tube transport may lead to a lower likelihood of pregnancy and impaired fertilization.

An ectopic pregnancy occurs when the fertilized egg implants and grows outside the uterine cavity, usually in the fallopian tubes. The pathophysiology of ectopic pregnancy has been linked to oxidative stress [169]. Numerous factors, such as infection, inflammation, and structural abnormalities, increase the susceptibility of the fallopian tube environment to oxidative stress. Elevated levels of ROS can damage the fallopian tube epithelium, impairing its normal function and disrupting the implantation process. The delicate balance of signaling molecules involved in the contact between the embryo and the fallopian tube wall may be disturbed by oxidative stress, increasing the likelihood of ectopic implantation. Ectopic pregnancy is a severe problem for the mother’s health and the woman’s fertility [12]. Implantation outside the uterus is associated with an increased risk of issues such as tubal rupture, bleeding, and loss of the affected fallopian tube. An ectopic pregnancy poses immediate dangers and has implications for future fertility. Surgical removal of the ectopic pregnancy or damaged fallopian tube can lead to adhesions and scarring of the fallopian tubes, further impairing their function and increasing the possibility of further ectopic pregnancies [170]. Managing oxidative stress is critical to maintaining good tubal function and reducing the risk of ectopic pregnancy.

### 5.8. Gestational Diabetes

Impaired glucose tolerance is a feature of the disease known as gestational diabetes mellitus (GDM), which occurs during pregnancy. New research suggests a possible link between oxidative stress and the development of GDM [171]. Insulin resistance, beta cell dysfunction and poor glucose metabolism during pregnancy have been linked to oxidative stress, defined as an imbalance between the formation of ROS and the antioxidant defense system. Insulin resistance, a significant aspect of GDM, has been linked to oxidative stress. Elevated ROS concentrations can cause oxidative damage to cells and disrupt insulin signaling pathways. Oxidative stress can lead to mitochondrial dysfunction, increase adipose tissue dysfunction, and activate proinflammatory signaling pathways. These signaling pathways interact to cause insulin resistance, which impairs glucose uptake and utilization in maternal tissues. GDM also exhibits beta cell dysfunction [172]. Pancreatic beta cell survival and function can be directly affected by oxidative stress. Elevated levels of ROS have been associated with the induction of apoptosis, disruption of insulin production and secretion, and oxidative damage to beta-cell components. Since insulin production is insufficient to meet the increasing demand in pregnancy, beta cell dysfunction contributes to hyperglycemia and the development of GDM. Because of the potential role of oxidative stress in the pathophysiology of GDM, antioxidant therapies have become increasingly popular as possible therapeutic approaches [173].

## 6. Possible Therapies to Treat the Harmful Effects of OS on the Female Reproductive System

Antioxidants, lifestyle modification, and drug interventions are possible therapies for OS-induced damage to the female reproductive system.

### 6.1. Lifestyle Changes

Lifestyle changes like a healthy diet, regular exercise, and stress management can protect the female reproductive system from the adverse effects of OS. Regular physical activity improves mitochondrial function and reduces OS levels, thus improving fertility. Antioxidants in diet, sourced from fruits and vegetables, can neutralize ROS and prevent cellular damage [174]. Body mass can significantly influence fertility, with both obesity and underweight conditions linked to hormonal abnormalities affecting fertility. Obesity can also lead to increased OS and inflammation [175]. Conversely, being underweight can result in irregular menstrual periods, difficulty ovulating, and higher OS levels, thus reducing a woman’s ability to conceive and carry a child to term. Hormonal and/or menstrual abnormalities caused by underweight and obesity seriously affect fertility [176]. Therefore, maintaining a healthy weight through a balanced diet and regular exercise is vital for supporting fertility and reproductive health. Specific health issues such as eating disorders or illnesses disrupting the ovulatory cycle can contribute to infertility and may require medical attention [177].

### 6.2. Antioxidants

Antioxidants scavenge reactive radicals to counteract the formation of ROS and promote the repair of oxidative damage to cell architecture [178]. Vitamins A, C, and E [179] and other natural chemicals such as polyphenols [180] are examples of non-enzymatic (supplemental/nutritional) antioxidants. In contrast, enzymatic (endogenous) antioxidants include, inter alia, glutathione, SOD, and glutathione peroxidase (GPx). Cofactors in enzymatic antioxidant systems, such as MnSOD and Se-GPX [4], are trace elements that include copper (Cu), zinc (Zn), manganese (Mn), and Se, most of which are consumed in the diet. Endogenous glutathione helps maintain exogenous antioxidants (vitamins A, C, and E) in their reduced form by neutralizing free radicals and ROS [26]. Throughout the female reproductive process, the antioxidant balance must be maintained and the harmful effects of ROS must be stabilized, which is why the dietary intake of exogenous antioxidants is crucial [181] (Figure 2).

Different mechanisms of action have been postulated for supplemental antioxidants. Improved endometrial blood flow, decreased hyperandrogenism, decreased insulin resistance, fertile cervical mucus, and an effect on prostaglandin production and steroidogenesis [182,183,184] all boost a woman’s ability to conceive. A review of the available scientific literature found that antioxidant intake improves oocyte and follicle quality, increases embryo implantation rates, and reduces pregnancy loss in women undergoing assisted reproductive technology [185,186,187]. The ideal dose and duration of antioxidant supplementation are not yet known, and further research is needed to determine the most effective treatment plan.

Antioxidants can destroy ROS while protecting cells, proteins, and DNA from oxidative damage. Several studies have linked OS to female infertility and have moreover shown that supplementation with antioxidants can increase the chances of pregnancy [186]. Of note, vitamin E, a fat-soluble antioxidant, has the capacity to protect cell membranes from oxidative damage. Pregnancy rates in women using ART therapies have shown improvement with vitamin E supplementation [188]. Vitamin C can neutralize ROS and restore vitamin E because it is a water-soluble antioxidant. Fertility in women with PCOS has been shown to improve when they take vitamin C supplements [189]. Glutathione peroxidase and other antioxidant enzymes cannot function adequately without Se, and, notably, it has been observed that the pregnancy rate in infertile women increases when they take Se supplements [190].

### 6.3. Medical Interventions

Numerous medical therapies seek to deal with the issue of OS-induced damage to the female reproductive system. Natural cycle IVF, minimal stimulation IVF, as well as the use of anti-inflammatory and antioxidant drugs like metformin and melatonin, are all procedures aimed at ovarian stimulation that reduce OS [191,192]. However, the full scope of benefits and potential risks associated with these therapies are not yet completely understood, necessitating further research for a more comprehensive investigation.

### 6.4. Assisted Reproductive Technology

Since the introduction of assisted reproductive technology (ART) several decades ago, thousands of infertile couples have been able to conceive children, a major advance for humanity. ART can be a beneficial recourse for couples grappling with various fertility issues. These include female infertility stemming from conditions like endometriosis or tubal infertility, male infertility, or fertility problems with causes that remain undetermined [193]. Intrauterine insemination, In vitro fertilization, and ICSI are examples of such methods. There are women who sidestep ART and merely take antioxidant supplements to increase their fertility, while others follow both paths simultaneously. Among the antioxidants that play a major role in infertility are the following: Coenzyme Q10 (CoQ10) is a powerful antioxidant required for the body to produce energy. It is shown to increase ovarian reserve, oocyte quality, and ovulation rates in women who undergo reproductive treatments [194]. The fat-soluble antioxidant vitamin E which, as mentioned, aids in the prevention of oxidative cell damage, has been shown among women using fertility treatments to boost the woman’s chances of getting pregnant as well as reduce the risk of miscarriage [195]. However, it remains uncertain whether it can help women with unexplained subfertility to increase their fertility.

## 7. Conclusions

OS, arising from an imbalance between ROS overproduction and the body’s antioxidant defenses, can be triggered by age and diseases impacting female reproduction. Conditions like endothelial dysfunction, early or recurrent pregnancy loss, IUGR, high blood pressure, premature birth, ectopic pregnancy and gestational diabetes are linked to OS. Adverse lifestyle habits and environmental pollutants may worsen OS, hinting at a potential role for antioxidants in mitigating these effects and improving fertility. However, current evidence regarding the effectiveness of antioxidants is inconclusive. Future research should prioritize understanding the role of antioxidants in managing female infertility, in order to deepen our knowledge and develop more effective treatments.

## Figures and Tables

**Figure 1 antioxidants-12-01490-f001:**
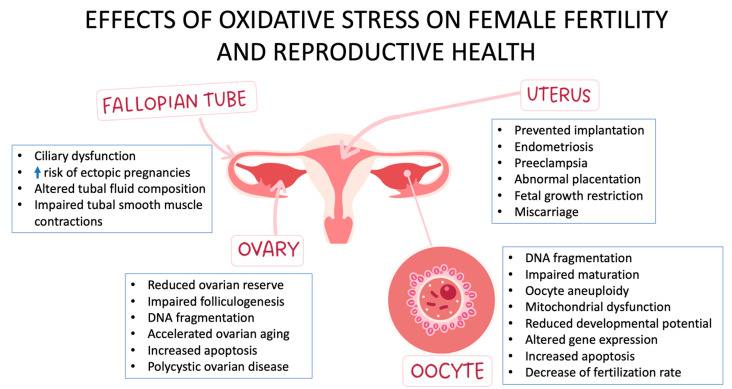
Impacts of Oxidative Stress on Female Fertility and Reproductive Health.

**Figure 2 antioxidants-12-01490-f002:**
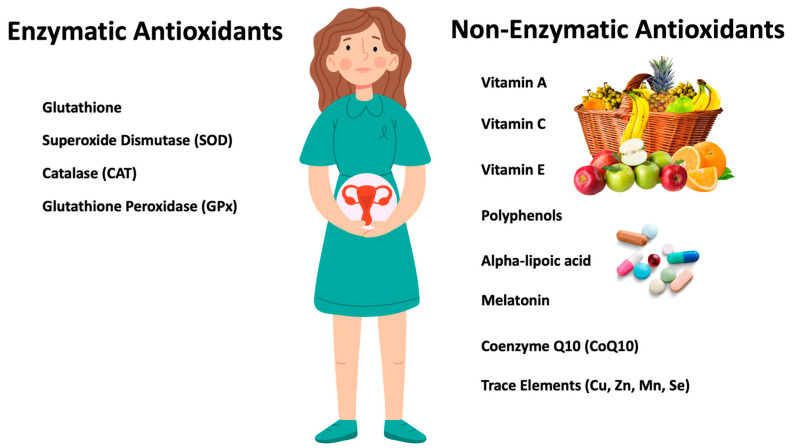
Antioxidants in Female Infertility Treatment.

## Data Availability

Not applicable.

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
