# Peer review of "The Silent Threat to Women’s Fertility: Uncovering the Devastating Effects of Oxidative Stress"

_antioxidants, 2023, doi:10.3390/antiox12081490_

Round 1

Reviewer 1 Report

Manuscript ID: 2506050, Title:   The Silent Threat to Women’s Fertility: Uncovering the  Devastating Effects of Oxidative Stress

 General comments

 The study presents original information on an interesting and very important topic of women’s reproductive medicine. Although the work is interesting and extensively reviews the literature regarding the association between OS and female infertility, it needs future revisions to improve the presentation of data. It is suggested that the authors review specific comments, which I hope will be helpful to the authors. Some parts of the manuscript are too extensive, and presentation information is unnecessary for the oxidative stress topic. I suggest reducing or deleting them. My specific comments are listed below.

 Specific comments

  Keywords

  The last section of the manuscript contains an evaluation of treatment options, please add some keywords about it,  e.g. “ antioxidants”. You can remove “female infertility”, it is not necessary when you give specific reproductive disorders.

 2.2. The Physiological Role of OS in Female Reproduction

 L 93-94: “ Other important factors…” rewrite this sentence, please.

L114-117: “Increased levels…”, delete this sentence. This is not a physiological condition description.

3.2. Body weight

L182-184: Expand the information about obesity and the irregular cycle or disturbed hormonal regulation. You can add some data from   6.1. Lifestyle changes section (which is too extensive).

3.3. Lifestyle Factors/  iii) Recreational Drug usage: Tetrahydrocannabinol

L 198-206:  Add additional information about oocytes  and THC connections and this citation:

URL=https://www.frontiersin.org/articles/10.3389/ftox.2021.647918            

DOI=10.3389/ftox.2021.647918 

 3.4. Environmental and Occupational exposures/ ii) Endocrine-Disrupting Chemicals

L215-221: What about women reproduction tack and phthalate; and some description

and citation Caporossi L, Viganò P, Paci E, Capanna S, Alteri A, Campo G, Pigini D, De Rosa M, Tranfo G, Papaleo B. Female Reproductive Health and Exposure to Phthalates and Bisphenol A: A Cross Sectional Study. Toxics. 2021 Nov 11;9(11):299. doi: 10.3390/toxics9110299. PMID: 34822691; PMCID: PMC8622554.

 4.1. Polycystic Ovarian Syndrome

L 226: replace please “inability to produce eggs,” with “inability to normal folliculogenesis”

L228: “anovulatory infertility”

L245-25: delete this information or put it into “6. Possible Therapies to Treat chapter”. The possible PCOS therapy is more suit for that section.

 4.2. Endometriosis

L 287-295 : „ A vital step…”. It is about therapy, replace this information with “6.2. Antioxidants” section.

 4.3. Unexplained Infertility

L 134-324: delete or reduce this information about folate metabolism or add to 6.2. Antioxidants” section.

 6.1. Lifestyle changes

L614-643: try to reduce that section is too extensive and has lots of general information about lifestyle changes. Try to highlight only the main factors which have involvement in OS therapy.

 6.4. Assisted Reproductive Technology

L 717-722: delete the sentences, that are redundant for the topic.

L726-2727: rephrase this sentence, something is missing there

 7. Conclusions

The conclusions are too long. Try to highlight the most important information about OS influence and therapy. It is not necessary to repeat in detail the previously described information.

e.g.:

 L750 -752: it is redundant

L758-769: it is better suited to “6.4 section”, not to conclusions.

Author Response

Dear Reviewer,

We deeply appreciate your thorough and constructive review of our manuscript. Your insightful comments are extremely valuable in refining and improving the presentation of our work. We have thoroughly considered and addressed all the points you raised, which have greatly helped us to enhance the clarity and coherence of the paper.

  1. We have updated the keywords to include "antioxidants", and removed "female infertility". We agree with your suggestion that mentioning specific reproductive disorders would make the scope of the paper more focused. Page 1, Line 33.
  2. In Section 2.2, we have rewritten the sentence on line 93-94 as suggested. Additionally, we have removed the sentence on line 114-117 as it did not contribute to the physiological context.
    Page 2, Lines 94-96.
  3. In Section 3.2, we have expanded the information about obesity and its connection to the irregular cycle and disturbed hormonal regulation. As suggested, relevant information from Section 6.1 has been integrated into this section. Page 4, Lines 190-203.
  4. In Section 3.3, we have added additional information about the relationship between oocytes and THC, and have included the recommended citation. Page 5, Lines 233-238.
  5. In Section 3.4, we have elaborated on the impacts of phthalates on women's reproductive health, and have referenced the suggested article. Page 5, Lines 254-258.
  6. For Section 4.1, we have replaced "inability to produce eggs" with "inability to normal folliculogenesis". Furthermore, the information about possible PCOS therapy has been moved to the appropriate section. Page 5, Line 263.
  7. For Section 4.2, we have moved the information about therapy to Section 6.2 as suggested. Page 6.
  8. In Section 4.3, we have trimmed the information about folate metabolism and moved the remaining content to Section 6.2. Page 7.
  9. We have condensed Section 6.1 to focus on the main factors relevant to oxidative stress therapy. Page 13, Lines 757-769.
  10. In Section 6.4, we have removed redundant sentences and have rephrased the sentence on line 726-727 for clarity. Page 14, Lines 974-976.
  11. We have streamlined the conclusions to focus on the most crucial aspects of OS influence and therapy. Redundant details have been removed and pertinent information has been shifted to Section 6.4. Page 15, Lines 1029-1037.

We have made every effort to ensure that the changes adhere to your suggestions. We believe that these revisions have significantly improved our manuscript and are hopeful that it will meet your approval. Thank you again for your time and helpful comments.

Sincerely,

Aris Kaltsas, MD, Phd

Reviewer 2 Report

Dear authors, this is a complete review of the state of art of oxidative stress and reproductive health. Nevertheless, the article is too long and some paragraphs are repetitive. Therefore, shortening and focusing would be highly appreciated by the reader. As examples (not exhaustive) on such situation are lines 245-258, 556-575.

Additionally, other minor checks are:

-Line 195: SOD is mentioned for first time without full transcription

-Lines 260.268: cervix is definitely not the most frequent location for endometriotic lesions; slowed menstrual cycle (whatever slowed means as this is not a recognized medical terminology) neither a predisposing factor.

-Lines 305-307: please revise the definition given for unexplained infertility as it is incorrect or the way it is expressed is misleading.

-Lines 305-324: it should be more clearly stated that MTHFR path anomaly is just ONE of the plausible explanations behind unexplained infertility. Peritoneal endometriosis, subtle tubal lesions, chronic endometritis… being more plausible.

-Line 425-426: This sentence is misleading. Please explain how ovarian stimulation is involved in all preeclampsia arising from spontaneous pregnancies.

-Line 473: FGFR is mentioned for first time without full transcription and not repeated again in the article, please delete to avoid redundancy.

-Lines 517-520: four main disease channels (?) are mentioned without description, followed by examples like uterine hypertrophy or cervical illness (whatever these mean as these are not recognized medical terminologies).

-Line 634 and following ones: please express in Kg.

-Line 722-723: please revise this sentence.

Author Response

Dear Reviewer,

We sincerely appreciate the time and effort you have spent reviewing our manuscript. We believe your comments have greatly helped us to enhance the quality of the paper. We have considered each point you have made and made the corresponding changes as detailed below:

  1. We have taken your suggestion to heart and significantly reduced the length of our manuscript, ensuring that our content is concise and non-repetitive.
  2. We apologize for the oversight and have included the full form of SOD (Superoxide Dismutase) at its first mention on line 195. Page 5, Line 222.
  3. We appreciate your insight regarding endometriotic lesions and the menstrual cycle. We have corrected these inaccuracies on lines 260-268, ensuring that the text aligns with recognized medical terminology and knowledge. Page 6, Lines 285-292.
  4. On lines 305-307, we have revised our definition of unexplained infertility to be more precise and less potentially misleading. Page 7, Lines 389-400.
  5. We have updated the information on lines 305-324 to clearly state that MTHFR path anomaly is only one of many plausible explanations behind unexplained infertility, including peritoneal endometriosis, subtle tubal lesions, and chronic endometritis, which are indeed more likely causes. Page 7, Lines 389-400.
  6. We have rephrased the sentence on lines 425-426 to clarify the relationship between ovarian stimulation and the development of preeclampsia in spontaneous pregnancies.
    Page 9, Lines 544-551.
  7. Following your advice, we have removed the reference to FGFR on line 473 to prevent redundancy. Page 10, Line 605.
  8. We have provided detailed descriptions for the four disease channels mentioned on lines 517-520. We've also removed terms like "uterine hypertrophy" and "cervical illness" that are not recognized in medical terminologies. Page 11, Lines 651-658.
  9. We appreciate your advice regarding the weight references. As per the suggestions of Reviewer 1, we have decided to delete these sentences instead of converting them to kilograms. Page 13, Lines 756-769.
  10. We have revised the sentence on lines 722-723 for better clarity and readability.
    Page 14, Lines 966-970.

We hope that these revisions address your concerns effectively. We believe the manuscript has been improved significantly as a result of your insightful feedback.

Thank you once again for your constructive comments.

Sincerely,

Aris Kaltsas, MD, PhD

Round 2

Reviewer 1 Report

The Authors adequately addressed my questions in the revised manuscript. I accept the present version of the manuscript.